# Temporal Evolution of Vapor Pressure Deficit Observed in Six Locations of Different Brazilian Ecosystems and Its Relationship with Micrometeorological Variables

**Rafael da Silva Palácios** [1,*], **Sérgio Roberto de Paulo** [2], **Iramaia Jorge Cabral de Paulo** [2], **Francisco de Almeida Lobo** [2], **Daniela de Oliveira Maionchi** [2], **Haline Josefa Araujo da Silva** [2], **Ian Maxime Cordeiro Barros da Silva** [2], **João Basso Marques** [2], **Marcelo Sacardi Biudes** [2], **Higo José Dalmagro** [3], **Thiago Rangel Rodrigues** [4] and **Leone Francisco Amorim Curado** [2,*]

[1] Instituto de Geociências, Universidade Federal do Pará (UFPA), Belém 66075-110, PA, Brazil
[2] Programa de Pós Graduação em Física Ambiental, Instituto de Física, Universidade Federal de Mato Grosso (UFMT), Cuiabá 78060-900, MT, Brazil; sergioufmt@gmail.com (S.R.d.P.); iramaiaj@gmail.com (I.J.C.d.P.); fdealobo@gmail.com (F.d.A.L.); dmaionchi@fisica.ufmt.br (D.d.O.M.); haline.jas@fisica.ufmt.br (H.J.A.d.S.); ianbarros@outlook.com (I.M.C.B.d.S.); jbassofisico@gmail.com (J.B.M.); marcelo@fisica.ufmt.br (M.S.B.)
[3] Programa de Pós-Graduação em Ciências Ambientais, Universidade de Cuiabá (UNIC), Cuiabá 78060-900, MT, Brazil; higojdalmagro@gmail.com
[4] Laboratório de Ciências Atmosféricas (LCA), Universidade Federal de Mato Grosso do Sul (UFMS), Campo Grande 79070-900, MS, Brazil; thiago.r.rodrigues@ufms.br
* Correspondence: rafael.pgfa@gmail.com (R.d.S.P.); leone.curado@fisica.ufmt.br (L.F.A.C.)

**Abstract:** In this study, data collected from 2000 to 2019 on vapor pressure deficit ($VPD$) and its relationship with micrometeorological variables (fire occurrences, aerosol concentration, temperature, and carbon flux) were analyzed in six locations situated in different Brazilian ecosystems: Rio Branco, AC; Manaus, AM; Alta Floresta, MT (within the Amazon Rainforest); Baia das Pedras, MT (Pantanal); Fazenda Miranda, MT (Cerrado); and Petrolina, PE (northeastern semiarid region). Temporal series analysis of $VPD$ was conducted by determining the principal component of singular spectrum analysis (SSA) for this variable in all locations. It was observed that the main component of SSA for $VPD$ is sensitive to local land-use changes, while no evidence of large-scale influences related to global climate change was observed. A strong coupling between $VPD$ values and local maximum temperature with monthly fire occurrence and logarithmic aerosol concentration profiles was also observed. The results of the study are discussed in the context of the ecosystems' carbon sequestration capacity. The combined results of the study indicate a scenario in which local land-use changes can compromise the capacity of Brazilian ecosystems to absorb carbon.

**Keywords:** vapor pressure deficit; singular spectrum analysis; land-use changes; microclimatology; global climate change

## 1. Introduction

Drought events have been extensively studied due to the natural hazards they pose, as variations in dry periods can strongly impact ecosystems. Although many researchers have assessed these risks, the processes and mechanisms involved in the duration and intensity of droughts are still not fully understood [1,2]. The period classified as a drought is characterized by above-average water deficit in ecosystems. This effect has ecological impacts and also affects the social dynamics of a region [3,4], and even a country.

Several studies [5–7] have indicated that drought events and extreme droughts are likely to intensify in the Amazon region due, among other factors, to anthropogenic activities. As a result, regions already experiencing long dry periods such as Cerrado and Pantanal (south of the Amazon Basin) are expected to have more frequent extreme drought events. These findings align with [8], who demonstrated an increasing vapor pressure

deficit ($VPD$) and intensification of drought periods in the Amazon region, as observed in the extreme droughts of 2005, 2010, and 2015.

Unlike other extreme weather events, drought involves changes in interconnected variables and may not be attributed to a single factor [9]. Particularly, the Amazon Basin region has experienced severe climatic anomalies [10–14] due to extreme drought events, such as those in 2005 and 2010, which led to the region's most negative annual carbon balance on record [15,16]. The recurring droughts in the Amazon between 2005 and 2016 have slowed down the hydrological system recovery and increased the risk of forest fires and tree mortality in the region [17,18]. These events, coupled with biomass burning emissions and land-use change, have fueled and accelerated climate change feedbacks [8,19,20].

In this scenario, an important factor in determining the duration of drought periods is the threshold value of atmospheric water demand [21,22]. The atmospheric water demand of an ecosystem is characterized by the vapor pressure deficit ($VPD$), which is the difference between the saturation vapor pressure and the actual pressure [23,24]. As saturation pressure is temperature-dependent, with the same amount of water vapor in the atmosphere, an increase in temperature leads to higher $VPD$ and lower relative humidity. Recent studies [22,25–30] have shown that the impacts of $VPD$ can be significant in natural ecosystems or agricultural crops, including crop water stress, crop yield reduction, tower fluxes, stomatal conductance, and aerodynamics affecting $CO_2$ and water fluxes. Especially in wetter ecosystems, stomatal conductance may be more sensitive to $VPD$ variations than soil moisture [28,31]. A high sensitivity of NDVI to $VPD$ variations was reported by [24].

In the studies conducted by [32,33], a significant decline in the apparent quantum yield ($\alpha$) and light response of net ecosystem exchange (NEE) parameters was found when subjected to high $VPD$ values. In another more recent study, [34], it was discovered that high $VPD$ values negatively impacted stomatal conductance, inhibiting the apparent quantum yield ($\alpha$) and daytime ecosystem respiration (Rd), thereby affecting photosynthesis. It was demonstrated in [8] that the tropical region of Latin America, particularly the Amazon Basin, is undergoing atmospheric drying with different regional patterns, and the atmospheric demand of forests is increasing, especially during drought periods.

The objective of this study was to analyze the impacts of drought through vapor pressure deficit ($VPD$) on changes in climate variable patterns, establishing a correlation with fire events in six locations in Brazil, five of which are part of the legal Amazon. The main goal was to establish a correlation between $VPD$ and the number of fire outbreaks. Additionally, an aim was to observe trends in the annual increase or decrease in the $VPD$ time series in the locations. Finally, an objective was to establish a relationship between $VPD$ and daily maximum temperature with carbon flux and relevant variables related to aerosol concentration, such as insolation index, aerosol optical depth ($AOD$), and black carbon concentration.

## 2. Materials and Methods

### 2.1. Study Area

The study was conducted at six micrometeorological sites in tropical regions, with five belonging to the Amazon Basin and one located in Northeast Brazil (Figure 1).

AERONET Site Information Database:

- Alta Floresta (AF-MT), Brazil (9.90835° S, 56.06439° W): The instrument is installed on top of the building of the Federal Institute of Mato Grosso (IFMT), which is located on the outskirts of Alta Floresta, close to an important highway in a region experiencing constant real estate growth. The main source of aerosols is forest fires. The instrument is well installed and has a 360-degree unobstructed view (https://aeronet.gsfc.nasa.gov/, accessed on 16 February 2023). For this site, a total of 4380 data points were used.
- Manaus (MA-AM) EMBRAPA, Brazil (2.89053° S, 59.96978° W): The instrument is installed approximately 30 km north of Manaus at EMBRAPA. It is located on top of a 15 m water box (https://aeronet.gsfc.nasa.gov/, accessed on 16 February 2023). For this site, a total of 2580 data points were used.

- Petrolina SONDA (Pet-PE), Brazil (9.06910° S, 40.32011° W): The Petrolina station is located in the Brazilian Northeast semi-arid region, which experiences high solar radiation. This region also has long drought periods and occasional heavy rainfall, strongly affected by El Niño and La Niña systems. The site is away from urban areas, situated in the EMBRAPA Agricultural Research Center for semi-arid climate. There is also a network of seven radio-linked meteorological stations around the radiometric station, managed by the local site manager (https://aeronet.gsfc.nasa.gov/, accessed on 16 February 2023). For this site, a total of 1410 data points were used.
- Rio Branco (RB-AC), Brazil (9.95747° S, 67.86935° W): This site is located on top of a 22 m tower in Rio Branco, Brazil (https://aeronet.gsfc.nasa.gov/, accessed on 16 February 2023). For this site, a total of 2940 data points were used.
- Fazenda Miranda (FM-MT), Brazil (15°17′ S, 56°06′ W): Located in the Cuiabá municipality. The vegetation is grass-dominated with sparse trees and shrubs, known as "campo sujo" or "dirty field" Cerrado [35]. For this site, a total of 1770 data points were used.
- Baía das Pedras Park (BP-MT), Brazil: Situated within the Private Natural Heritage Reserve—RPPN SESC (16°39′ S, 56°47′ W)—in the Poconé municipality, bordering the municipality of Barão de Melgaço (about 160 km from the capital Cuiaba). This region is part of the Pantanal, considered one of the largest sedimentation plains on Earth [36,37]. For this site, a total of 1080 data points were used.

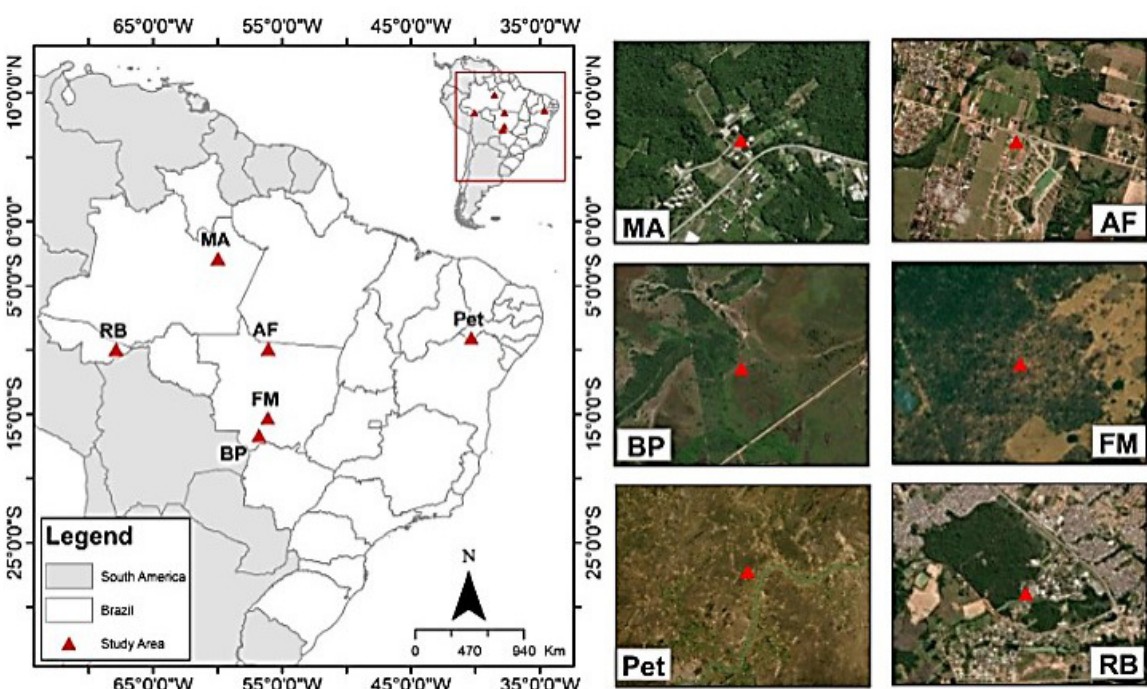

**Figure 1.** Localization of the towers in the study area.

## 2.2. Micrometeorological Measurements

The National Institute of Meteorology—INMET (https://portal.inmet.gov.br/, accessed on 17 February 2023)—provided the micrometeorological data for the following sites: Alta Floresta (AF-MT) from 2002 to 2017, Manaus (MA-AM) from 2011 to 2018, Rio Branco (RB-AC) from 2000 to 2009, and Petrolina (Pet-PE) from 2004 to 2016. For the Baía das Pedras (BP-MT) site from 2017 to 2019 and FM-MT (Fazenda Miranda) site from 2009 to 2015, data from micrometeorological towers installed at those locations were used.

The AERONET network (https://aeronet.gsfc.nasa.gov/new_web/networks.html, accessed on 17 February 2023) provided the aerosol optical depth (*AOD*) data. The measurements of particulate matter at the BP site were conducted using an aethalometer (Model AE33). The aethalometer is a device that provides real-time optical measurement of the

concentration of black carbon (*BC*) (a soot aerosol resulting from combustion) present in the atmosphere. The air sample is extracted through the inlet port at a flow rate of 2.0 L per minute using a small internal pump. The flow rate is monitored by an internal mass flow measurer and electronically stabilized at the setpoint value entered in the software. The device collects the sample on a quartz fiber filter tape and performs continuous optical analysis while the sample is being collected. The aethalometer measurements were corrected according to the methodologies of [38,39] with correction parameters specific to [40].

The data on fire hotspots were provided by the National Institute for Space Research—INPE (https://queimadas.dgi.inpe.br/queimadas/portal, accessed on 17 February 2023). The measurements of $CO_2$ flux were conducted using the eddy covariance system during two different periods: from January 2014 to July 2015 and from September 2016 to June 2017. These measurements were analyzed in conjunction with photosynthetically active radiation (PAR) measurements, allowing for the evaluation of $CO_2$ flux variations as a function of PAR. The eddy covariance system included a 3D sonic anemometer used to measure three-dimensional orthogonal components of velocity ($u_x$, $u_y$, $u_z$) and capture high-frequency sonic temperature fluctuations. In addition, infrared gas analyzers were employed to measure $CO_2$ concentrations. Between 1 January 2014 and 30 June 2014, a WindMaster 3D anemometer (Gill Instruments Ltd., Lymington, UK) with an accuracy of $\pm 0.05$ m/s was utilized. From 1 January 2015 to 30 July 2017, an R.M. Young 3D anemometer (Model 81000, R.M. Young Company, Traverse City, MI) with an accuracy of $\pm 0.05$ m/s was employed. For the measurement of $CO_2$ concentrations, an open path infrared gas analyzer LI-7500A (LI-COR Biosciences, Lincoln, NE) with an accuracy of 1% was used [41].

The *VPD*, in kPa, was calculated using the following equation:

$$VPD = e_s - e, \tag{1}$$

where $e_s$ is the saturation vapor pressure of moist air, in kPa, Furthermore, $e$ is the actual vapor pressure given by,

$$e_s = 0.61078 \cdot 10^{7.5T/(237.3+T)}, \tag{2}$$

$$e = UR \cdot e_s / 100, \tag{3}$$

where $T$ is the air temperature in $^\circ$C and $UR$ is the relative humidity of the air in %.

### 2.3. Singular Spectrum Analysis

The data were qualitatively compared by analyzing the obtained time series. In order to identify possible systematic trends in the daily average *VPD* over the study period, a principal component analysis was performed using the singular spectrum analysis (SSA) method, following the procedures described in [42], for each of the study locations. This method is capable of extracting the main trend from the data by identifying the various orthogonal trends (components) that are mixed in the time series data, as well as measuring the power (eigenvalue) of each trend. Thus, it is possible to eliminate less important trends that may be attributed to disturbances or stochastic elements.

## 3. Results and Discussion

### 3.1. Temporal Variation of *VPD*

According to Figure 2, there is a correspondence between the monthly average *VPD* and the number of monthly fire hotspots recorded in the corresponding location, especially in forest (MA, RB), savannah and Pantanal regions (FM, BP). Even considering that the numerical values of the relative maxima of the number of fire outbreaks and *VPD* may not be the same in all locations, with periods where *VPD* is relatively high even with a low number of fire outbreaks, the overall feature of the two variables corresponds in the sense that they vary over time with maxima and minima at corresponding time instances. This indicates that in these regions, prolonged dry periods can lead to restrictive conditions for ecosystem functioning, as evidenced by the increase in the number of fire hotspots during

periods of maximum $VPD$. Therefore, a significant coupling between the number of fire hotspots and $VPD$ is observed in all studied ecosystems.

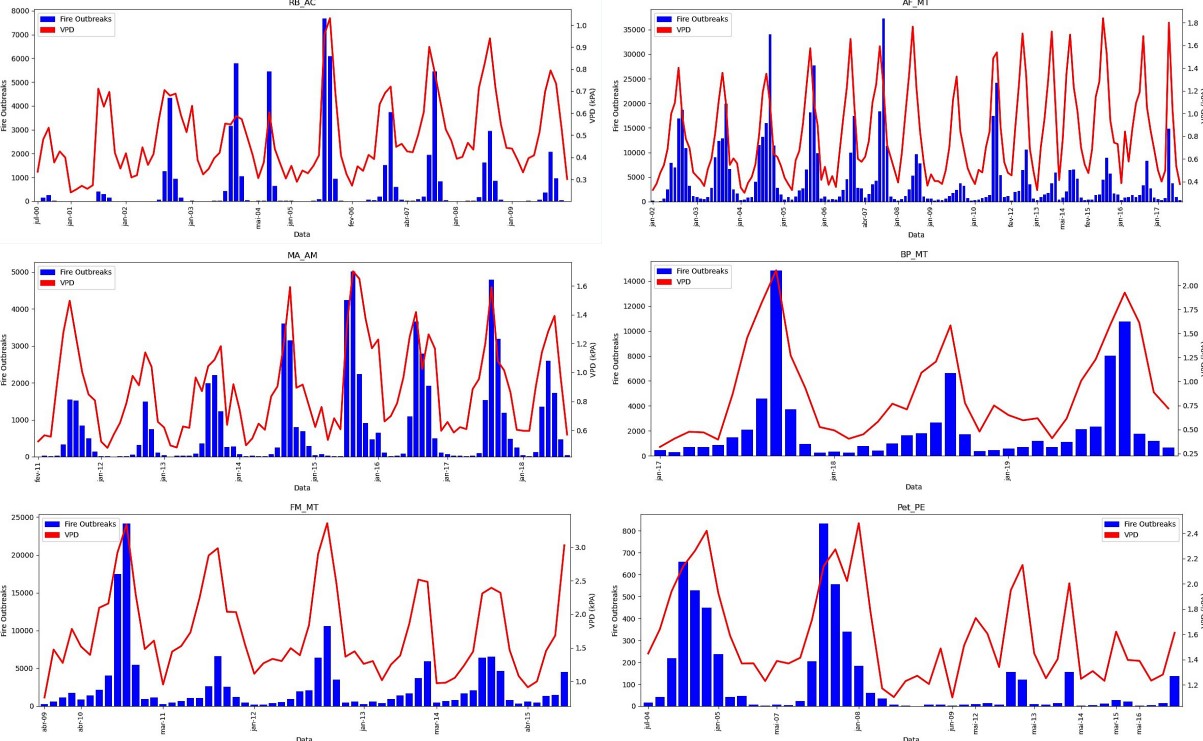

**Figure 2.** Interannual variation of $VPD$ and number of fire outbreaks ($FQs$) during the studied period in different ecosystems. On the *x*-axis, the values correspond to a date specified by the month and year.

Since $VPD$ can serve as an efficient indicative index of such restrictive conditions, possible long-term changes (whether due to global climate change or local alterations) in these conditions can be characterized by trends observed in $VPD$ data. Such changes may indicate modifications in the dry season duration of these regions and in the dynamics of net ecosystem exchange, as discussed by [43], as increased $VPD$ can affect photosynthesis through stomatal closure [34,44].

The SSA analysis was performed for the data from all the study sites, and the results are presented in Figures 3 and 4. Through the analysis of SSA, the main trend of the $VPD$ time series was determined. The SSA method calculates the main trends of a time series by generating an autocorrelation matrix whose elements are obtained through the correlation between the original time series and the time-lagged series. The eigenvalues of this matrix correspond to the weight of each principal component of the described phenomenon, while the eigenvectors are used to calculate the principal components. Figures 3 and 4 present the eigenvalue spectra and the first principal component of the $VPD$ data for each location. The eigenvalue spectrum displays the value of each eigenvalue on a logarithmic scale in descending order. It is observed that the first eigenvalue in all cases is at least one order of magnitude higher than the others. This indicates that only one principal component reflects the data trend, while the others can be considered perturbations or noise. Additionally, these figures show the principal component of each location, which corresponds to the main trend of the $VPD$ time series. Therefore, it is possible to identify in which locations the $VPD$ shows an increasing or decreasing trend.

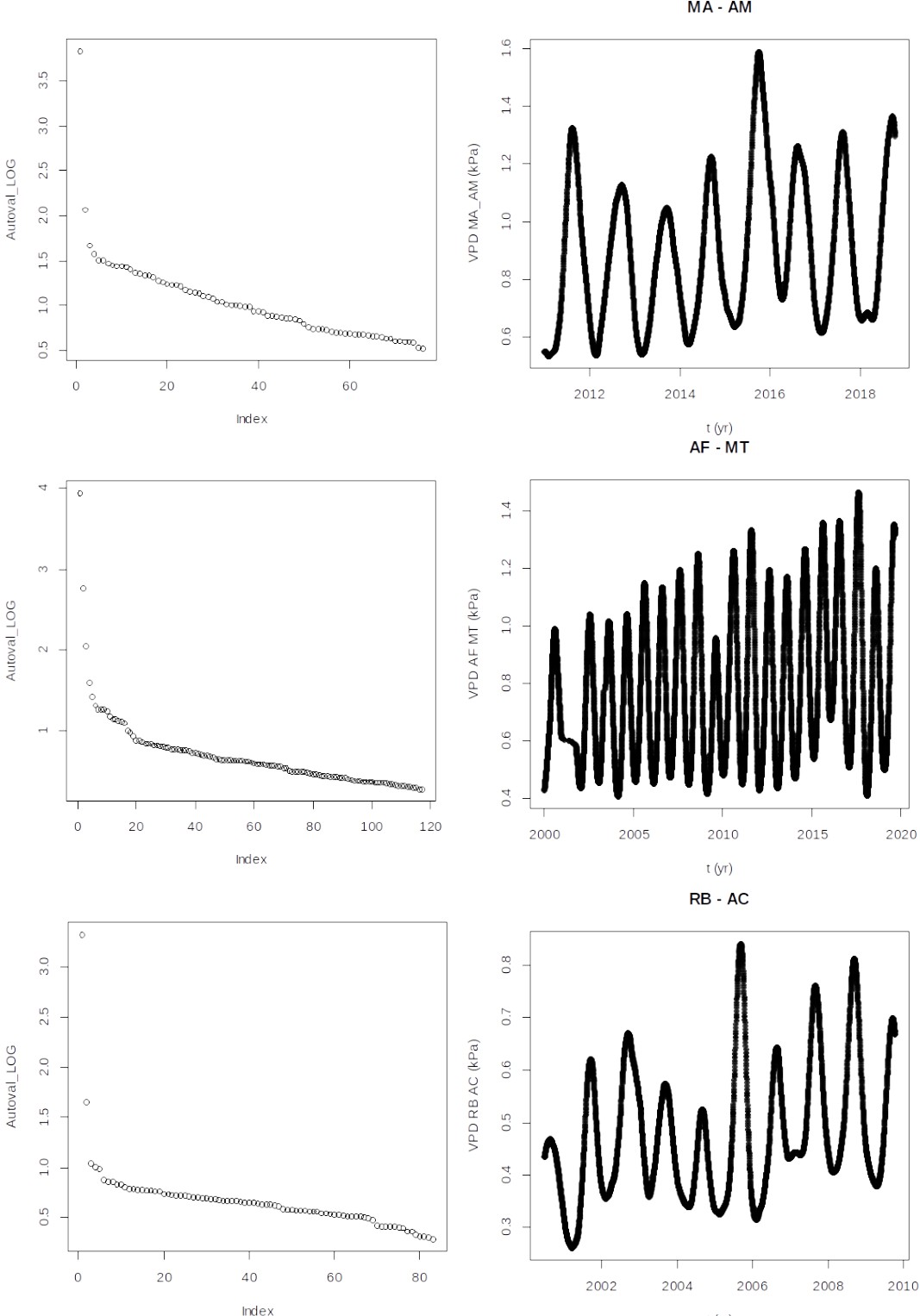

**Figure 3.** Principal component analysis of *VPD*. On the left, eigenvalue spectrum in logarithmic scale. On the right, the principal component of *VPD* in Manaus (MA) (**top**), Alta Floresta (AF) (**middle**), and Rio Branco (RB-AC) (**bottom**).

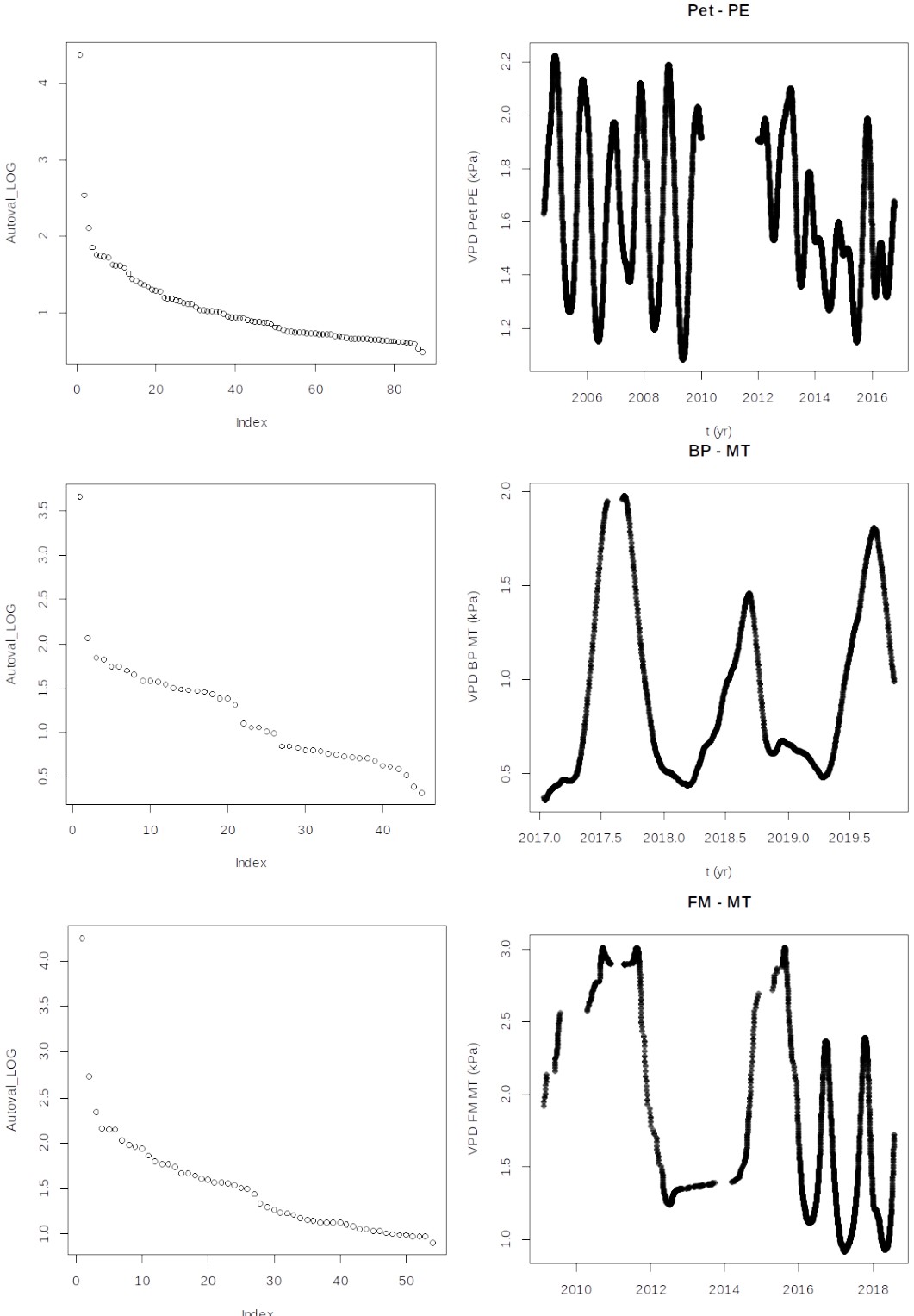

**Figure 4.** Principal component analysis of $VPD$. Left: eigenvalue spectrum in logarithmic scale. Right: the principal component of $VPD$ in Petrolina, PE (**top**); Baia das Pedras (BP) (**middle**); and Fazenda Miranda (FM) (**bottom**).

With respect to the profile of the studied component, different behaviors are observed regarding the temporal evolution of $VPD$. No significant variation in the $VPD$ behavior is observed in Manaus or Baia das Pedras, with both systems having a large amount of available water. On the other hand, in Rio Branco and especially in Alta Floresta, the

main component indicates a consistent increase in the annual maximum value of $VPD$. In Fazenda Miranda and Petrolina, on the other hand, a decrease is observed. In the latter location, there is a decrease in the annual variation amplitude of $VPD$, which is the difference between the maximum and minimum annual values. This behavior is opposite to that of Alta Floresta (AF), where an increase in amplitude is observed.

It is important to note that this description is conditioned by the periods of data availability. For Baía das Pedras (BP) in the Pantanal, for example, there are only 3 years of available data. However, the observed profile of the first principal component is consistent with the behavior of the ecosystem, which is characterized by irregular alternation between drier and wetter years [41].

In the case of Petrolina, the principal component analysis indicates a systematic decrease in the maximum value of $VPD$ during the period 2004–2016. This behavior may be related to the increase in irrigation projects in the vicinity of Petrolina, which has contributed to a statistically significant decrease in the flow of the São Francisco River in the region during the same period [45].

A systematic decrease is also indicated by the analysis of the first principal component of Fazenda Miranda (FM). Although the available data may not be sufficient for long-term inferences, the observed decrease in Figure 4 is consistent with the fact that the study site corresponds to an abandoned pasture where natural vegetation is continuously growing.

An opposite trend, namely an observable increase in the principal component of $VPD$, is notable in the data from Rio Branco (RB-AC) and Alta Floresta (AF). In the case of Alta Floresta, during the studied period (2002–2017), there was a significant change in land use. Data describing such changes is presented in [46], wich analyses the temporal series of temperature, humidity, and precipitation in the region between 1986 and 2015. A systematic increase in average and maximum temperatures, as well as a systematic decrease in humidity and precipitation, was observed, all of which can be attributed to deforestation caused by agribusiness. As indicated by Figure 3, these land-use modifications particularly affected the annual maximum value of $VPD$ rather than the minimum, indicating an increase in the severity of dry periods.

Regarding the increase in $VPD$ in Rio Branco (as indicated by Figure 3, showing an increase in both the annual maximum and minimum values), [47] presents a detailed study on the changes in land use that have occurred in the municipality. Since the 1970s, there has been intense urbanization and population growth, resulting in an expansion of the urban area. Results obtained from satellite images (INMET) indicate a systematic increase in local temperature between 1990 and 2010 [48]. The linear regression analysis of the data show a two-degree increase in temperature during that period. They also present data based on satellite images consistent with a systematic increase in albedo, which is a reflection of urbanization.

Based on these results, it can be inferred that the principal component analysis of $VPD$ using SSA yields consistent results with local land-use changes, indicating that this method is sensitive to such alterations and can be used as an indicator. Furthermore, the data presented do not indicate a global or at least continental influence, suggesting that within the scope of the studied locations, local modifications impose more significant changes in $VPD$ than non-local factors such as El Niño.

However, it should be noted that the above results stem from the trends presented in the individually considered time series. When comparing the absolute mean values of $VPD$ across different study locations, as well as their amplitude, it is observed that the locations where there is a systematic increase in $VPD$ are not necessarily the ones with the highest average value of this variable. Table 1 presents the minimum, maximum, and mean values of $VPD$ for all sites over the studied years, along with the standard deviation of the daily means.

**Table 1.** Minimum, maximum, and mean values of $VPD$ (kPa) in each studied location.

| Site | Min | Max | Avg | Std |
|------|------|------|------|------|
| AC-RB | 0.241 | 1.031 | 0.492 | 0.265 |
| AF | 0.302 | 1.839 | 0.828 | 0.471 |
| AM-MA | 0.478 | 1.700 | 0.897 | 0.465 |
| BP | 0.318 | 2.156 | 0.869 | 0.664 |
| FM | 0.759 | 3.359 | 1.743 | 0.964 |
| PE | 1.099 | 2.479 | 1.591 | 0.542 |

The data in Table 1 are consistent with the hypothesis that locations with nearby forest vegetation and, consequently, greater water availability exhibit greater resistance to $VPD$ variations. This is evident in the case of AC-RB, which had the lowest mean and standard deviation, as well as the lowest maximum $VPD$ value throughout the study period. Although there is a trend of increase, the mean, maximum, and minimum $VPD$ values are significantly lower than those of other locations. Similarly, Manaus and Alta Floresta, which are located within the Amazon Rainforest, had lower absolute $VPD$ values compared to Baia das Pedras, Fazenda Miranda, and Petrolina. Therefore, locations situated in regions with less dense vegetation, such as Cerrado and Pantanal, exhibited the highest standard deviations (9.64 hPa for FM and 6.64 hPa for BP, respectively), along with PE, located in the northeastern semiarid region of Brazil. This indicates that in a scenario of land-use change and deforestation, these regions can be drastically affected by an increase in $VPD$.

*3.2. Changes in Environmental Factors Associated with the Increase in $VPD$ and Wildfire Occurrences*

The increase in $VPD$ values and wildfires can lead to negative effects on ecosystems, resulting in increased temperature [23,24] and the presence of black carbon in the atmosphere of these regions, which can positively contribute to the trend of solar radiation received during the late dry season, leading to an increase in daily maximum temperature [7]. The following figures present the impacts of $VPD$ and fire hotspots on environmental variables at the six study sites.

Figure 5 presents a quantitative analysis of the relationship between $VPD$ and the number of fire outbreaks. A logarithmic fitting between the two variables is displayed. Similarly, in Figure 6, an increasing logarithmic fit is found between the number of fire outbreaks and the maximum temperature.

In Figure 5, it can be observed that there is a positive correlation between the number of fire hotspots and the $VPD$ value at all six sites, indicating a coupling between these variables, especially in forest regions like Manaus, which show significant $R^2$ values—AM ($R^2 = 0.7457$); PE ($R^2 = 0.7598$); BP ($R^2 = 0.8143$); FM ($R^2 = 0.7764$)—indicating that these biomes may be more sensitive to environmental changes due to fire activity. As discussed by [19], the effects of black carbon aerosols released by biomass burning in the Amazon and savannah regions are of significant importance for understanding local and regional climate changes.

The data collected at all research sites suggest a logarithmic–exponential relationship between $VPD$ and fire hotspots, where $VPD$ saturates for a high number of fire hotspots, and, conversely, it is highly sensitive to changes in the number of fire hotspots for a low number of this variable. One possible explanation for this relationship is that during dry periods when $VPD$ reaches high values, the ecosystem conditions are highly favorable for fires, leading to a high and variable number of fire hotspots. On the other hand, for less dry conditions, $VPD$ shows high sensitivity to the number of fire hotspots, suggesting that vapor pressure deficit can be an important index for evaluating fire risk.

Indeed, there is no universality when it comes to the maximum values of $VPD$ as they depend on the specific ecosystem in which they occur, as previously mentioned. Therefore,

the maximum value of locally measured $VPD$ seems to depend on non-local conditions, such as the ecosystem in which the location is situated.

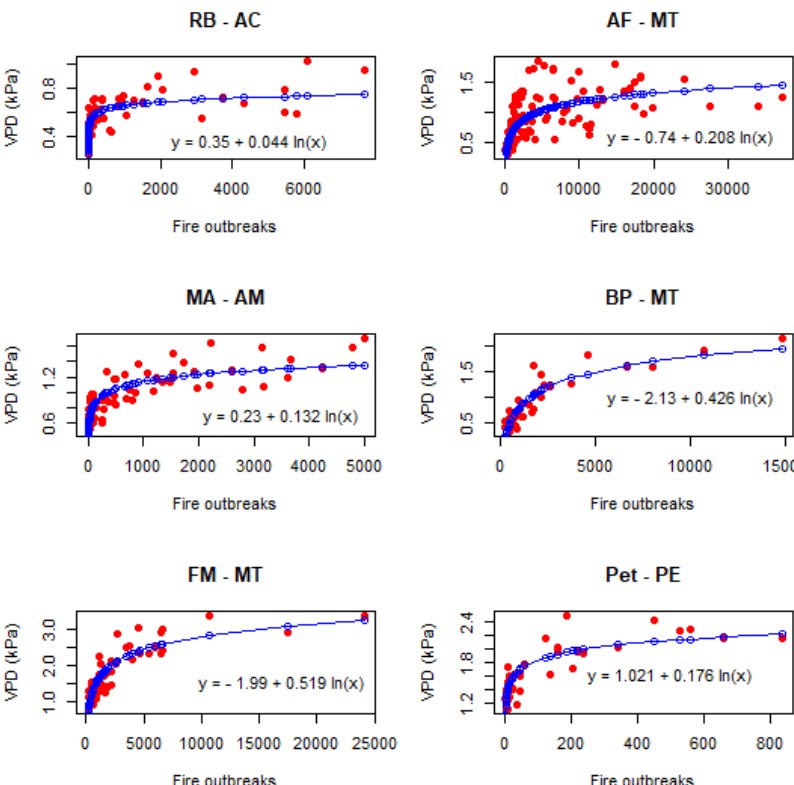

**Figure 5.** Relationship between $VPD$ and number of fire outbreaks in the studied locations.

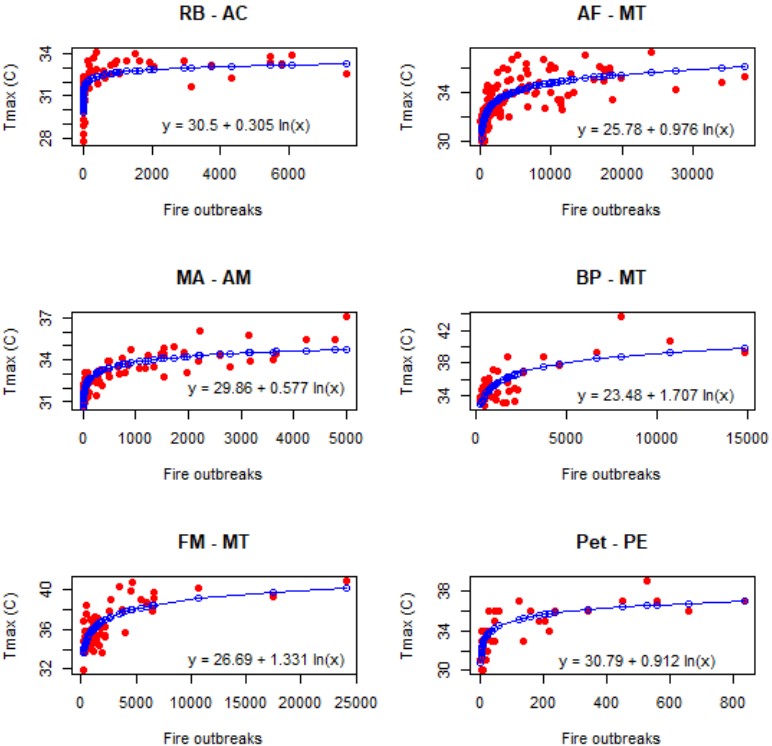

**Figure 6.** Relationship between monthly accumulated fire outbreaks ($FQs$) and monthly average maximum temperature in the studied locations.

The similar pattern observed in Figure 5 can also be seen in Figure 6, which depicts the relationship between $VPD$ data and maximum temperature measured at each location. In all research sites, higher maximum temperatures were associated with a greater number of wildfires, following a logarithmic–exponential pattern. This indicates a strong tendency towards changes in the thermal patterns of these regions. At least part of this trend can be attributed to aerosols. The increase in solar radiation received due to the presence of aerosols from more solar radiation-absorbing wildfires can contribute to the higher maximum temperatures [7]. As these aerosols re-emit absorbed radiation in the infrared spectrum, they can play a significant role in increasing maximum temperatures in the tropical zone of South America, confirming the results discussed here. It is worth noting the southern regions of the Amazon Basin, predominantly covered by savanna and the Pantanal (FM, BP), which recorded maximum temperatures above 40 °C when there was an increase in wildfires. This indicates that these regions are more sensitive to climate change and may experience more drastic changes due to human activities, potentially leading to longer dry periods in these tropical regions [3].

### 3.3. Effects of Wildfire Aerosols and Insolation Ratio on *VPD* and Other Environmental Factors in the Amazon Basin

Except for Petrolina, due to the absence of data, an attempt was made to establish a relationship between local maximum temperature ($Tmax$) and $VPD$ with a variable that reflects the concentration of aerosols, depending on data availability. In the case of Rio Branco (AC) and Fazenda Miranda (FM), aerosol optical depth ($AOD$) measured at 870 nm was used. In Alta Floresta (AF) and Manaus (AM), the sunshine ratio was utilized, and in the case of Baia das Pedras, black carbon ($BC$) concentration was considered.

In all cases, attempts were made to establish a logarithmic relationship, as shown in Figures 5 and 6. However, for Manaus and Alta Floresta, where sunshine ratio data ($n/N$) were used, Figure 7 shows that a linear relationship fit the data better. It is possible that during the study periods, the values of $n/N$ did not reach small enough values to observe a substantial increase in $Tmax$ or $VPD$ with the sunshine ratio, which would be the typical profile of a logarithmic relationship.

Indeed, it is observed that the aerosols from fires ($AOD$ or $BC$) and the sunshine ratio ($n/N$) are positively correlated with $VPD$ and maximum temperature, indicating that these agents may be coupled to changes in environmental components. The increase in wildfires in the Amazon region can have dramatic effects on the ecosystems within the Amazon Basin, both in forested regions and in other ecosystems. It is worth noting these effects in the sites of Manaus (AM) and Baía das Pedras-Pantanal (BP), where the coefficients of determination ($R^2$) had higher values during the study period, 0.6009 and 0.609 for maximum temperature, and 0.7415 and 0.8248 for $VPD$, respectively. These results are in line with [3,7], which report that the presence of aerosols from fires reduces cloud cover and increases maximum temperature, as well as alters cloud albedo [49]. These effects of aerosols from fires in forest and savannah regions of the Amazon Basin can already be observed. A strong correlation between aerosol absorption index and daily maximum temperature in India is also reported in [50].

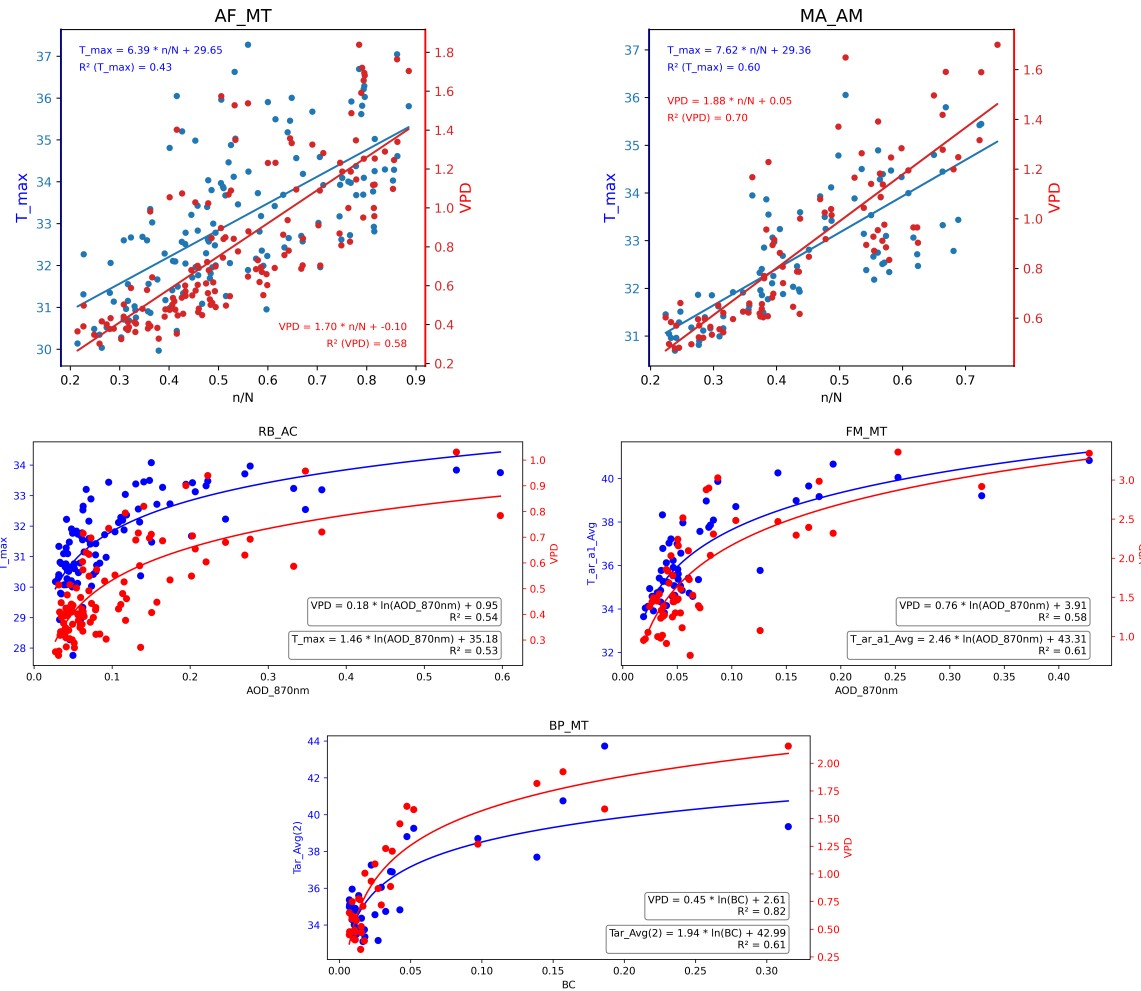

**Figure 7.** Relation between local maximum temperature (blue dots) and DPV (red dots), and variables that reflect the concentration of aerosols in the air.

### 3.4. Implications of $VPD$ Changes on Carbon Flux and Photosynthesis: A Case Study

The implications of $VPD$ changes on carbon flux in the Pantanal wetland were also analyzed. This analysis was possible at this site because carbon flux data were available from December 2013 to June 2017, measured by an eddy covariance system installed during that period. Figure 8 shows the relationships between monthly average carbon flux and both $VPD$ and vapor pressure (e).

The figures above show the linear dependence of photosynthesis (negative $CO_2$ flux) on vapor pressure and $VPD$. As the air becomes drier, resulting in decreased vapor pressure and an increase in $VPD$, it imposes water-restrictive conditions for photosynthesis, as outlined in the literature. These conditions are characterized by limited water access for vegetation and water stress, leading to a significant decrease in carbon uptake by the ecosystem. Several studies have investigated the negative influences of increased $VPD$ on carbon uptake by vegetation, such as [28,31], which report that stomatal conductance is more sensitive to $VPD$ increase than soil moisture, especially in wetter areas. Besides that, high $VPD$ values negatively impact stomatal conductance, inhibiting apparent quantum yield and daytime ecosystem respiration, thereby affecting photosynthesis [34].

In the conditions of Baia das Pedras, for example, every 1 hPa increase in $VPD$ corresponds to a decrease of 0.27 μmol/m²s in the carbon uptake rate. This implies that if this relationship holds true in other locations, significant restrictions in carbon uptake

capacity can be expected in areas where there are significant land-use changes, as observed in this study.

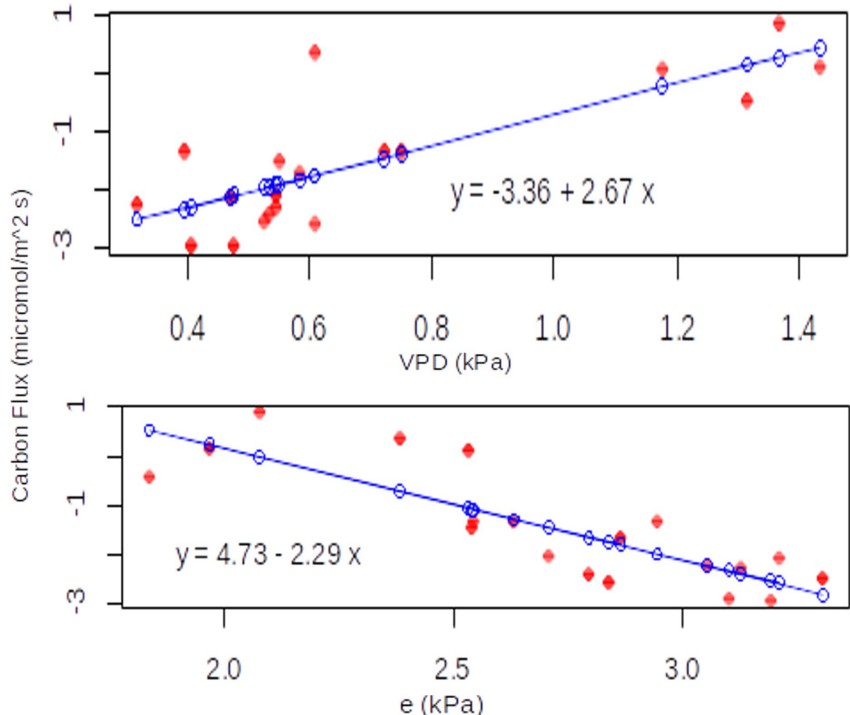

**Figure 8.** Relationship between monthly mean carbon flux and *VPD* (**top**) and vapor pressure (**bottom**) observed at Baia das Pedras site (Pantanal).

The results of this study, as a whole, indicate certain relationships between the variables studied in various locations under conditions of high vapor pressure deficit values. In summary, the main findings were the existence of a positive relationship between both the vapor pressure deficit (*VPD*) and maximum daily temperature with the number of fire outbreaks. This relationship implies a high susceptibility to fire occurrence for specific temperature values (above 32 °C) and *VPD* values (>1 kPa). Under these conditions, a significant aerosol emission process resulting from burning is expected in all studied locations. The increase in aerosol concentration, which also exhibits a logarithmic relationship with temperature according to the results of the case study conducted at the experimental site of Baia das Pedras, is directly linked to the ecosystems' ability to absorb $CO_2$, as a linear relationship was found between the monthly average carbon flux and both *VPD* and water vapor pressure (ranging from 2 to 3 $\mu$mol/m$^2$s per kPa). It is important to note that the linear relationship corresponds to a wide range of *VPD* variation.

The SSA analysis demonstrated that scenarios of increasing or decreasing *VPD* values from year to year are associated with local changes involving anthropogenic activities. A decrease in *VPD* was observed in an area with intensified extensive irrigation and local vegetation restoration, while increases in *VPD* were observed in areas with intensified urbanization and deforestation for livestock farming. These results indicate a scenario in which the lack of regulation of environmental preservation policies can jeopardize ecosystems' carbon absorption capacity.

The results obtained in this study can contribute to a deeper understanding of the susceptibility of Brazilian tropical ecosystems to drought conditions, characterized by significant vapor pressure deficit values. They can aid in shaping the probable future scenario for these ecosystems, considering that land-use change associated with deforestation can definitively contribute to an increase in vapor pressure deficit and, consequently, the loss of ecosystems' carbon absorption capacity.

## 4. Conclusions

An analysis was conducted to examine the relationship between $VPD$ and daily maximum temperature, considering parameters associated with the presence of aerosols such as $AOD$, black carbon concentration, and carbon flux. However, due to data availability limitations, this analysis was only performed for the Baia das Pedras location. A logarithmic relationship was found between $VPD$ and aerosol variables, while a linear relationship was observed between $VPD$ and carbon flux.

The results obtained in this study indicate that land-use changes that have occurred in Brazil in recent decades have a significant impact on vapor pressure deficit ($VPD$), with important consequences for the local microclimate. In the research sites explored in this study, significant changes were observed in the temporal $VPD$ series, both in terms of systematic increases and decreases, depending on the local conditions. In areas where there is systematic deforestation and urbanization, such as Alta Floresta, MT, and Rio Branco, AC, a significant increase in $VPD$ values was observed. Conversely, in areas where there is increased irrigation, such as Petrolina, PE, and vegetation recovery, such as Fazenda Miranda in the Cerrado region of Mato Grosso, a systematic decrease in $VPD$ values was observed. In the six study sites, no significant influence of large-scale factors, such as those characterized by El Niño, was observed, as no common patterns were found in the temporal $VPD$ series.

The relationships obtained between $VPD$ and micrometeorological variables such as temperature, aerosol concentration, and carbon flux demonstrate a strong connection between these variables under different land-use conditions. This indicates that $VPD$ has the potential to be an efficient index for characterizing the microclimatic conditions of a locality.

This work will be continued through the expansion of the collected data, with the aim of reducing the uncertainties associated with the observed relationships between the studied variables.

**Author Contributions:** Conceptualization, R.d.S.P., S.R.d.P., H.J.D. and L.F.A.C.; Methodology, D.d.O.M.; Validation, L.F.A.C.; Investigation, F.d.A.L., J.B.M., M.S.B., T.R.R. and L.F.A.C.; Data curation, S.R.d.P., I.J.C.d.P., H.J.A.d.S., I.M.C.B.d.S. and T.R.R.; Writing—original draft, S.R.d.P., I.J.C.d.P., F.d.A.L. and J.B.M.; Writing—review & editing, R.d.S.P., D.d.O.M., M.S.B., H.J.D., T.R.R. and L.F.A.C. All authors have read and agreed to the published version of the manuscript.

**Funding:** This study was supported by the Pará Research Support Foundation, FAPESPA, project 2022/43638 and PROPESP/UFPA (PAPQ), process 23073.047927/2023-50.

**Data Availability Statement:** The AERONET website provides data analysis and dissemination tools at https://aeronet.gsfc.nasa.gov (accessed on 16 February 2023). Data can be viewed in charts using the data display interface, acquired using the data download tool, analyzed, and downloaded using some analysis tools provided by AERONET. The INPE website provides data analysis and dissemination tools at https://queimadas.dgi.inpe.br/queimadas/portal (accessed on 17 February 2023).

**Acknowledgments:** The authors would like to express their gratitude to Brazilian Coordenação de Aperfeiçoamento de Pessoal de Nível Superior (CAPES), Conselho Nacional de Desenvolvimento Científico e Tecnológico (CNPq), and Pró-Reitoria de Pesquisa e Pós-Graduação da Universidade Federal do Pará PROPESP/UFPA (PAPQ) for supporting this study, as well as the National Institute of Research in Pantanal (INPP) and the Large-Scale Biosphere-Atmosphere Program (LBA), coordinated by the National Institute for Amazonian Research (INPA), for the use and availability of data, logistical support, and infrastructure during field activities.

**Conflicts of Interest:** The authors declare no conflict of interest.

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
