# Peer review of "Temporal Evolution of Vapor Pressure Deficit Observed in Six Locations of Different Brazilian Ecosystems and Its Relationship with Micrometeorological Variables"

_forests, doi:10.3390/f14081543_

Round 1

Reviewer 1 Report

Dear Authors,

Please find my comments in the attached file.

Presently, your paper lacks the structure. Objectives not present, methods for each objective as well. Please take this issue seriously into account.

Regards

Author Response

Thank you for your feedback. All comments have been accepted and corrected according to the guidelines. Our corrections will be sufficient for the publication to be accepted.
The Objectives Has been rewritten, the methods has been explained.

Reviewer 2 Report

The article “Temporal evolution of vapor pressure deficit observed in six locations of different Brazilian ecosystems and its relationship with micrometeorological variables” by Rafael da Silva Palacios et. al. contains an analysis of the dependence of the relationship of vapor pressure deficit on micrometeorological variables  in six locations in different Brazilian ecosystems.

I think that the material of the article is interesting and important for the environment.

The authors need to make some corrections.

1)      Line 132: (Am-Ma, Ac-Rb), - unexplained abbreviation;

2)      please, indicate how were the eigenvalues of the corresponding principal components drawn in Figures 3-4 calculated and also it is necessary to explain what this behavior of eigenvalues means;

3)      the inscriptions in Figures 2-4 are very small;

4)      pictures 3-4 are not done neatly;

5)      in the captions under Figures 5,6, 8 please indicate what does each color mean.

Author Response

Thank you for your feedback. All comments have been accepted and corrected according to the guidelines. Our corrections will be sufficient for the publication to be accepted.
The abbreviation Has been explained in study area in Material and Methods topic (lines 72-96).
Has been included the explanation in Material and Methods topic (lines 135-142) and in Results and Discussions (lines 163-176).
The Figures has been improved.
Has been included the explanation in lines 282-285.

Reviewer 3 Report

I have read the manuscrip with title "Temporal evolution of vapor pressure deficit observed in six locations of different Brazilian ecosystems and its relationship with micrometeorological variables" which is interest and well written. I have moderate comments:

- Provide the innovation of the study.

- In discussion section add drawbacks and future works. Is essential for a manuscript

suggested literature

Simulating future groundwater recharge in coastal and inland catchments. Water Resource Management.

 - In the conclusion section provide the main results with 3-4 bullets.

Is fine

Author Response

Thank you for your feedback. All comments have been accepted and corrected according to the guidelines. Our corrections will be sufficient for the publication to be accepted.
Has been included the explanation in Results and Discussions (lines 163-176) and (339-358).
Has been included the explanation in Results and Discussions (lines 163-176) and (339-358).
Has been included the explanation in Conclusions (lines 360-365).

Round 2

Reviewer 1 Report

Dear authors,

Thank you for improving the manuscript. Some small issues still remains.

Regards

Author Response

Thank you for your feedback. All comments have been accepted and corrected according to the guidelines.